# Standardized methods for rearing a moth larva, *Manduca sexta*, in a laboratory setting

**Emma K. Spencer**[1]*, **Craig R. Miller**[1,2], **James J. Bull**[1,2]

**1** Department of Biological Sciences, University of Idaho, Moscow, Idaho, United States of America,
**2** Institute for Modeling Collaboration and Innovation, University of Idaho, Moscow, Idaho, United States of America

* espencer@uidaho.edu

## Abstract

The larval tobacco hornworm, *Manduca sexta*, has been used in a laboratory setting for physiological studies and for pathogen virulence studies. This moth offers a much larger size than the commonly used wax moth (*Galleria mellonella*), and it can thus be used for a greater variety of assays, such as repeated sampling of the same individual, growth measurements, and tissue sampling. Yet their occasional use in research has led to a minimally documented set of rearing methods. To facilitate further adoption of this insect model, we expanded on previously reported protocols and developed our own rearing methods, which we report here. Our protocol requires little specialized equipment, with a cost less than $100/month for the feeding and maintenance of a laboratory colony of about five hundred larvae of differing instar phases. The low cost generalized equipment and supplies, and the simplification of the standardized protocols allows for an easy entry point for rearing tobacco hornworm populations. We also describe a few methods that are relevant to the uses of these organisms as infection models.

## Introduction

Invertebrate organisms have been used as models in scientific research as early as the 19th century. The first publications in the late 1800s of using invertebrate organisms in research were focused on marine invertebrates [1]. In more modern times, the commonly used invertebrate species for research are fruit flies (*Drosophila melanogaster*) and nematode (*Caenorhabditis elegans*). These invertebrate species have led to some of the most groundbreaking scientific discoveries in the 21st century [2–6].

One invertebrate emerging in popularity as a model organism is the greater wax moth (*Galleria mellonella*). The first mention of greater wax moth in a scientific publication described genetic characteristics following exposure to X-rays in 1938 [7], but the species wasn't fully established as a widely recognized invertebrate model until

**Data availability statement:** All relevant data are within the paper and its Supporting Information files.

**Funding:** Research reported in this publication was supported by the National Institute of General Medical Sciences of the National Institutes of Health under Award Number P20GM104420. The content is solely the responsibility of the authors and does not necessarily represent the official views of the National Institutes of Health. The funders had no role in study design, data collection and analysis, decision to publish, or preparation of the manuscript.

**Competing interests:** The authors have declared that no competing interests exist.

the late 1960s [8,9]. Use of wax moth larvae in research has experienced a 10-fold increase in publication numbers in the past ten years [10].

Many invertebrates have practical benefits for laboratory research. Compared to vertebrates, they are inexpensive, have short generation times, can be kept in large population sizes, and raise fewer ethical issues. However, the commonly used invertebrates are physically small. Adult nematodes are only ~ 1 millimeter long [11]. *Drosophila* species are only a couple of millimeters long, and wax moth larvae range in size from 14–28 millimeters with an average weight of 280 mg [12,13]. For many purposes, a larger size and longevity is useful, especially for injections for virulence studies to establish chronic infections. Another advantage of a larger size is that other manipulations, such as sequential sampling, and tissue dissections, are obtainable.

An insect that exhibits the advantages of a large size is the tobacco hornworm moth (*Manduca sexta*). *M. sexta* is commonly known as a tobacco hawk moth as an adult, and the tobacco hornworm as a larva. This species is a member of the order Lepidoptera, and the larval form is a pest of plants in the Solanaceae family. It is unusual among model invertebrates in attaining a large size, with the final larval stage in the range of 10–13 grams – a size approaching that of a small mouse [14]. The tobacco hornworm has been used as a laboratory model in its larval form for studies of insect physiology, metamorphosis, and biochemistry [15–19]. The immune system and biological functions of the tobacco hornworm are also well characterized [14,20–24]. The genome and transcriptome information for *M. sexta* is also available. With advancements in sequencing technology, a group was able to generate a new chromosome level genome assembly named JHU_Msex_v1.0, which consists of 470 Mb with a scaffold N50 of 14.2 Mb [25]. The extensive body of research and the abundance of studies available underscore the tobacco hornworm's significance as a valuable and important model organism.

Hornworm larvae can be purchased from suppliers who rear them as food for captive reptiles. The cost per hornworm varies depending on the supplier, but generally they range from $0.40 to $1.00 per individual [26,27]. This cost is significantly lower per individual as compared to mice, which are between $12.50 – $40.00 per individual depending on the strain and size [28]. The price per hornworm does not include the expedited shipping costs required to ship live insects, and suppliers often limit the quantity of hornworms shipped from 12–25 total individuals. The hornworms are grouped for shipping by size, but generally the range in size and instar age is quite varied which makes standardization for studies difficult. The stress of shipping, including drastic temperature changes, jostling, and the pressure of housing all larvae in one container throughout the shipping process can lead to death, cannibalism, or reduction in tolerance to insults. The consistency of rearing and uniformity of survival and durability amongst purchased cohorts is unknown. Establishing a laboratory colony enables control over the model's consistency, size, age, and quality, but standardizing rearing methods are not obviously available for hornworms.

This paper provides standardized methods for the rearing of tobacco hornworms in a laboratory. The few published procedures are a useful starting point. Our protocols are designed for simplicity and uniformity, and provide many details absent in existing protocols.

## Materials and methods

### Strain source

Our first generation of moths was a mix obtained from two sources: 28 larvae from ReptileSupply.com (Wichita, KS), and 50 eggs from Great Lakes Hornworms (Bruce Township, MI). Both suppliers are based in the United States and specialize in producing larvae for reptile food. From this original population, more than 50 individual larvae were reared to produce subsequent generations. Additional individuals were supplemented into the laboratory population throughout the year in the form of 500 eggs sourced from Great Lakes Hornworms. It was apparent that these individuals were not obtained directly from the wild by the supplier, rather that there is a considerable (but unknown) degree of lab adaptation in the history of this stock (as noted below).

### Equipment and resources

The initial cost of essential items for rearing tobacco hornworms is low, making the entry point for starting a laboratory colony simple. Essential materials are listed in Table 1, and the basic steps of propagation are shown in Fig 1.

The initial investment of the materials listed in Table 1 is calculated to be about $626, which includes the multiple bags of powdered food necessary to feed a colony of larvae for a whole year. This cost estimate only includes the indicated items in Table 1 and doesn't account for the labor of feeding and performing maintenance for the colony.

**Artificial diet.** Once hornworms hatch from their eggs, they are put on an artificial wheatgerm-based diet [29]. The artificial diet used for the maintenance of this laboratory population is premixed from Great Lakes Hornworms; the recipe is proprietary, but is a pre-mixed dry powder composed of wheatgerm, soy, vitamins, and minerals.

The artificial diet is prepared by combining 800 mL of boiling water, 100 g of pre-mixed dry food, and ten grams of BD™ Difco™ agar. The mixture should be stirred thoroughly until all the dry ingredients are completely combined, before removing the mixture from the heat source and pouring into a container for storage. The diet is UV surface sterilized for 15+ minutes prior to feeding, and all instruments that touch and handle the food are sterilized with 70% ethanol.

### Protocol for rearing

The steps described in this section are outlined in Fig 1. Additional information on the stages within the life cycle is given in later sections. From hatching to pupation, the hornworms are kept at 28 °C and 30–60% relative humidity (RH).

1) Adult moths are housed in a butterfly enclosure from the moment they emerge until they die, up to 40 moths per enclosure. Mating occurs in this enclosure. Eggs are collected from this enclosure daily or every other day, depending on egg deposition rate. Eggs are typically laid on the undersides of items in the enclosure or on the sides of the netting. Eggs laid on cardboard or plastic surfaces may be carefully rolled in a circular motion with minimal finger pressure to

**Table 1. List of necessary materials and pricing to rear tobacco hornworms.** Materials and equipment used for rearing tobacco hornworm populations in a laboratory setting, and their approximate costs.

| Materials for larvae | Adult Moth Materials | Equipment |
|---|---|---|
| *Drosophila* flasks (177 mL) with cotton stoppers ($200 for 500) | Butterfly/moth enclosures (60.96 cm X 60.96 cm X 91.44 cm) ($30) | Incubator (25–37°C) |
| Substrate (coconut fiber, sawdust, or other alternative burrowing substrate for prepupal stage) (~$10) | Hummingbird feeders (~$12) | Hotplate or microwave to heat water for diet preparation. |
| Pre-mixed wheatgerm-based artificial diet sourced from Great Lakes Hornworms ($64/4 lb.) (Need three 4 lb bags per year if feeding 100 grams once a week) and BD™ Difco™ agar ($289/500 g) | Plastic gutter guard (~$7) | |

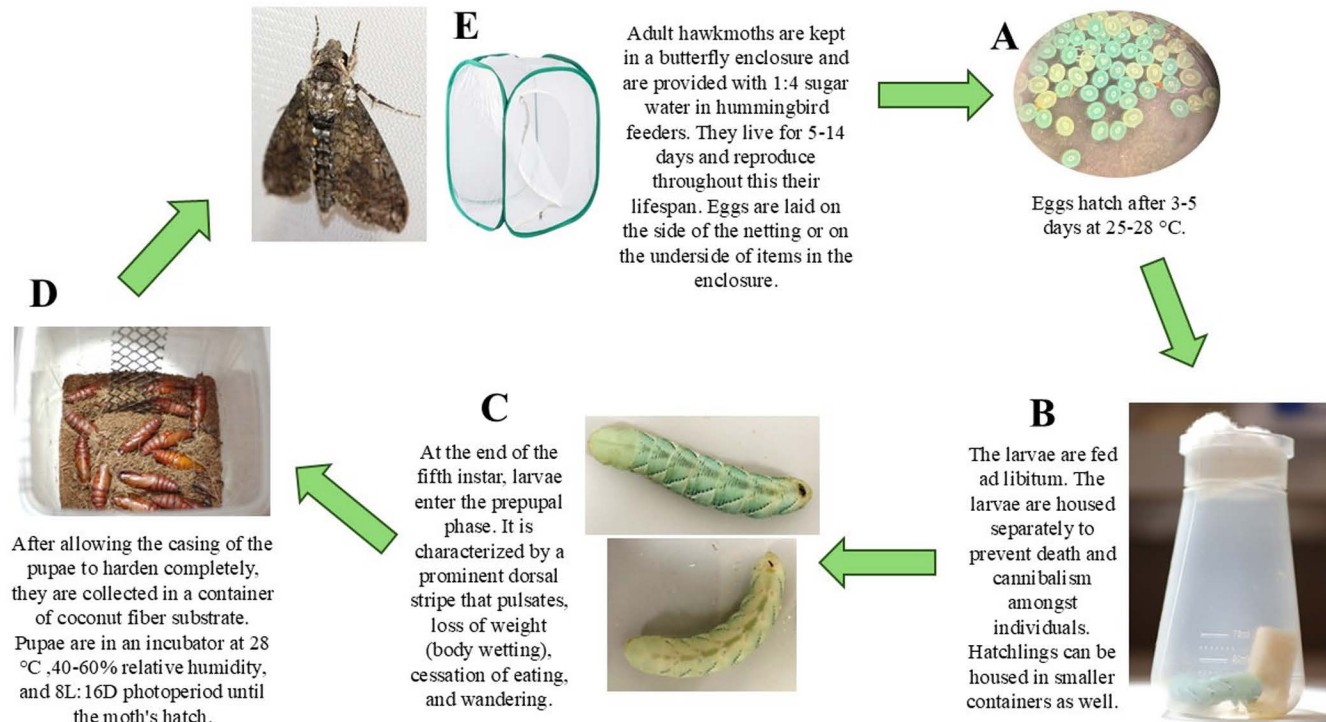

**Fig 1. Tobacco hornworm lifecycle with materials.** This figure shows the equipment outlined in Table 1. The major steps of the protocol are summarized above and visually display the major lifecycle phases. A) Photo of fertilized eggs that are two days post laying under a dissection microscope. B) A fifth instar larva in a plastic Drosophila flask. C) Two different prepupal larvae demonstrate the prominent dorsal stripe that characterizes this larval phase. D) A container with coconut fiber substrate with several week-old pupae that are tanned and hardened. The plastic gutter guard is in the container to allow for moths to hang and expand their wings immediately following pupal eclosion. E) An adult hawk moth and an example of the butterfly enclosure that they are housed in.

release the egg free of damage. At 25–28°C, 30–60% RH, fertilized eggs hatch 3–5 days after laying but may be briefly refrigerated to delay hatching (see below). Typically, the hatchlings ingest their egg casing before crawling upwards in search of food.

2) Transfer the hatchlings to individual *Drosophila* flasks with a small block of food at the bottom. The hatchlings are placed directly on the food to avoid starvation. One hatchling is housed in each flask. Smaller containers can be used to house the first several instars but are not essential. The larvae are housed individually to avoid cannibalism. Use flat forceps to handle hatchlings by the horn on their posterior. The horns at hatching are fragile and may break off during handling. A small amount of hemolymph may drain from the breakage site, but horn loss is rarely fatal.

3) Replace food as necessary for growing larvae; remove frass from the containers every two days. The larvae will grow from about 6.8 mg to about 10 g at the end of the fifth instar (Table 2), so the amount of food consumed increases with development. Use 1% bleach and 70% ethanol to clean and sterilize the containers, forceps, and other surfaces that the food and larvae encounter. The larvae shed and eat their skins at the end of the first four instars. The most common cause of larval death observed was molting problems with incomplete skin shedding. In low humidity conditions skins may stick to the larvae, but the remaining skin shed can often be coaxed off by wetting the worm with water and using forceps to remove the shed.

**Table 2. Descriptors for larval development by instar. Color and other visual differences amongst instars.**

| Larval Instar | Coloration | Additional developments |
|---|---|---|
| **Descriptors for larval development by instar** | | |
| Hatchling | Transparent | |
| First Instar | Blue pigment develops 12–48 hours after hatching | Posterior horn turns black |
| Second Instar | Pigmentation darkens but no stripes | Posterior horn turns red |
| Third Instar | Stripes develop and spots alongside abdominal segments begin to develop | |
| Fourth Instar | Stripes and abdominal spots more prominent | |
| Fifth Instar | Brighter and more prominent white and black stripes, with the abdominal spots a brighter yellow-orange color. | Black hoods called crochets are present on the tip of each proleg. |

4) Larvae transition to the prepupal phase at the end of the fifth instar; this stage is easily recognized (see below). Pre-pupal worms are placed in containers with 8 cm deep coconut fiber substrate, or other inert substrate, to allow them to burrow and complete pupation. If the pupae are left in the container until eclosion, plastic gutter guard, or other material are added to these containers as vertical climbing substrate for emerging adults (Fig 1 (D)).

5) Pupal containers are checked daily for adults. New moths are transferred to a net enclosure with other adult moths for mating and feeding. Adults are given a diet of 1:4 sugar water placed in miniature hummingbird feeders or vials. A female typically mates with one male. Once mated, the female begins oviposition [30].

**Temperature, humidity, and photoperiod.** Larvae were observed to survive and grow within the temperature range of 22–35 °C, but otherwise everything reported here was conducted at 25–28 °C, chiefly 28 °C. A suitable relative humidity (RH) was found to be 30–60%. Our early studies used a lower relative humidity (10–20%) but the hatchlings and first instar larvae died frequently from desiccation.

The larvae were housed in a dark incubator to maintain temperature, thus not exposed to a specific photoperiod. In contrast, adults were exposed to an 8L:16D photoperiod. The adult moths are held in a room with no windows, and the photoperiods are controlled by the fluorescent overhead lights. Adults began flying at the beginning of the dark phase, even in the small cages used here. Unmated females are reported to initiate 'calling' within the first two hours of the onset of dark [30]. The calling behavior is where the ovipositor is extended beyond the abdomen to expose the pheromone glands. We also observed this calling behavior in our adult stock. The period during which mating was observed lasted 3–4 hours, and oviposition by the females began during the following dark period [30]. We observed females depositing eggs during the dark phases as well as the first thirty minutes of the light period as well.

## Results

### Biology of the life cycle in captivity

**Basics.** The life cycle consists of an egg stage, five larval stages known as instars, which are marked by skin shed between each instar, a prepupal phase, a pupa, and the adult moth (Fig 2, Table 2). Most of the emphasis here is on larvae, as they are suited for many manipulations and have five distinct stages that may need to be distinguished for research purposes.

The total egg to egg duration of the life cycle is six-to-nine weeks. The larval life cycle from hatching to pupation spans 25 days (±3 days), with the five larval instars spanning 15–20 days, the prepupal stage five-to-seven days; pupation to eclosion spans two-to-five weeks. Adult hawk moths live five-to-ten days and can begin laying eggs as soon as eight hours after eclosion [30]. With the length of the life cycle, we could produce about nine generations/year.

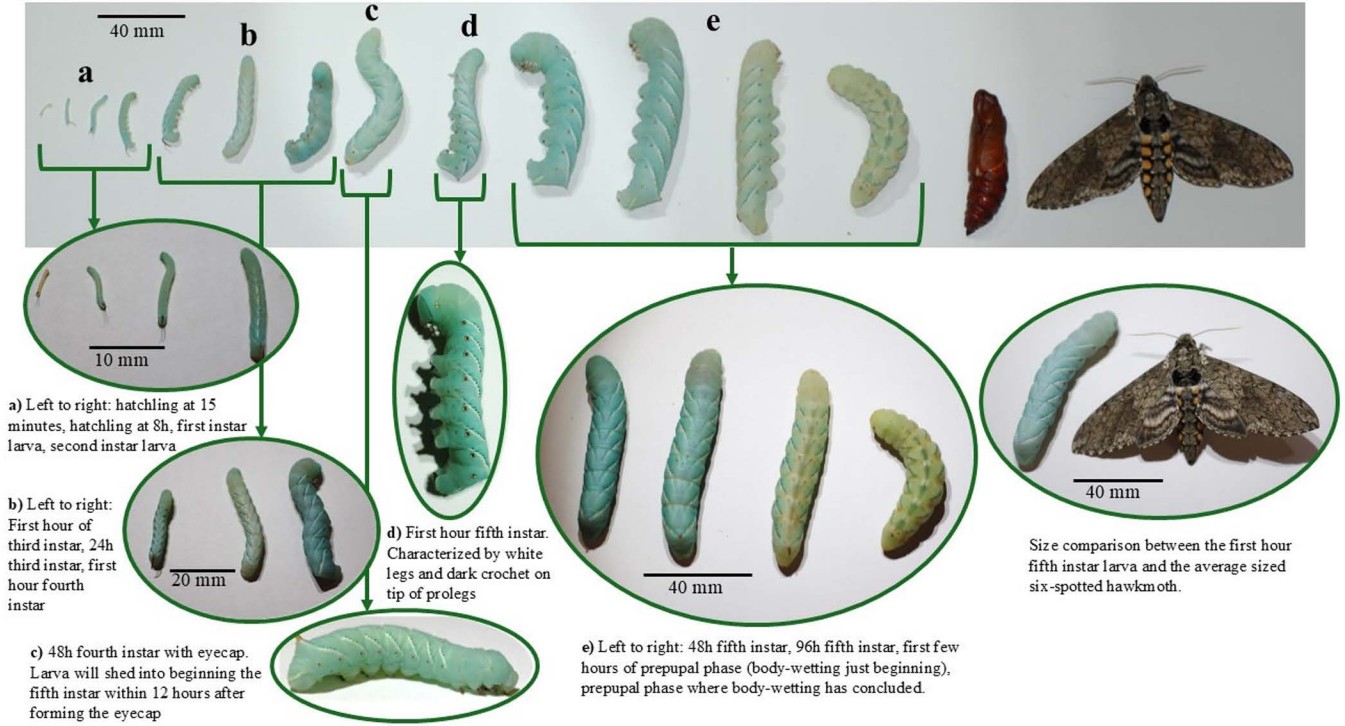

**Fig 2. The life stages of the tobacco hornworm.** The life stages of the tobacco hornworm from hatchling to adult, with an emphasis on the larval phases. The larvae pictured in groups a-e on the top of the figure are the same larvae but are different images from a higher magnification. The arrows indicate the higher magnification images that correspond to the same life stages from the life cycle above.

**Growth, mass, and duration of larval stages.** Basic characteristics of individual instars are reported in Table 3. The first larval instar begins 12–48 hours after hatching. Previous work and our observations here found that each of the first four instars ends with a molt sleep, a period of 14–25 hours in which the larvae does not feed, defecate, or move until ecdysis, or skin shedding [31]. The end of the fifth instar, however, is more gradual and is signaled by physiological indicators of pre-pupation, such as the cessation of feeding, body wetting, prominent dorsal pigmentation, and wandering behavior. The relationship between the mass and length throughout the larval phase is log-linear, as shown in Fig 3.

**Fifth instar measurements and manipulations suited to experimentation.** The most growth and development occur in the fifth instar larval phase, with larvae increasing in mass about ten-fold (Fig 4). The fifth instar is also the longest lasting of the larval phases (four-to-five days) (Table 3). These factors make this instar well suited for various

**Table 3. Size and duration of larval development by instar; sample sizes exceed 40.**

| Size and duration of larval development by instar | | | |
| --- | --- | --- | --- |
| **Larval instar** | **Duration** | **Mean mass** | **Length of instar** |
| First instar | 2–3 days | 6.8 mg +/- 1.8 mg | 6–10 mm |
| Second instar | 2–3 days | 20.6 mg +/- 8.8 mg | 10–15 mm |
| Third instar | 2–3 days | 99.0 mg +/- 8.5 mg | 16–22 mm |
| Fourth instar | 3–4 days | 0.76 g +/- 0.5g | 25–40 mm |
| Beginning of Fifth instar | 4–5 days | 1.5g +/- 0.85g | 45–60 mm |
| End of Fifth instar | | 8.25g +/- 1.5g | 75–90 mm |

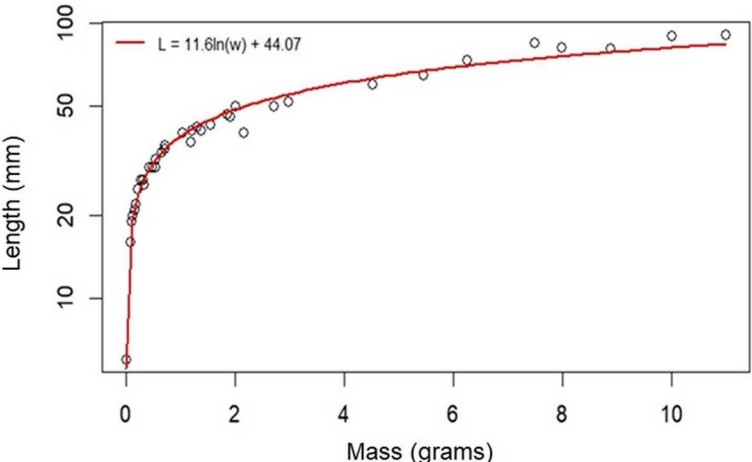

**Fig 3. Relationship between mass and length during larval development.** Each point represents an individual larva selected at different larval phases (hatchling – fifth instar). The red curve is fitted as a regression to log(mass) as the X variable, length as the Y variable.

experiments, such as hemolymph extraction, injection, tissue dissection, and frass collection. The relatively constant growth rate of the fifth instar (Fig 4) provides a phenotype beyond simple survival by which to assess experimental effects.

The hemolymph can be extracted every day without severe detriment to the larva. The larvae will die if too much hemolymph is drawn, or if the extraction site fails to coagulate and the larvae continues to bleed. The maximum volume to draw at a time without major disruption to the growth rate and health of the larva is 300 μL. This volume would most likely be excessive if done daily (Table 4). To avoid disruption to growth rate, a volume of 50–100 μL hemolymph can be drawn daily. When hemolymph is drawn, the larvae should be put on ice for 15 minutes prior. Fig 5 and Table 4 represent nine fifth instar larva that had hemolymph drawn up to five times over the course of six days.

A 30-gauge needle inserted between the seventh abdominal segment and terminal segment was found suitable to draw the hemolymph. Two of the nine larvae perished after the third draw, but the relative weight of the other larvae were compared to the average growth rate of a sample of fifth instar larva that received no treatment. In Fig 5, the blue line labeled 'No Treatment AVG' is the same population showing the average relative mass change over time as the red line in Fig 4. This blue line serves as the control for which to compare the change of mass of the individual larva. Generally, the relative mass change for larvae that received serial hemolymph draws is lower than that of the no treatment control. Table 4 shows the corresponding hornworm numbers and the total volume of hemolymph drawn, as well as the volume of hemolymph drawn each time.

Beyond single or serial hemolymph draws, fifth instar larvae can also be used for experiments that involve injections. For an injection, as opposed to hemolymph draws, an area that will cause the least amount of bleeding is optimal. We used the crease between abdominal sections between the third and fourth set of prolegs. An injection volume of 10–50 μL was used; larger volumes risk bleeding and a change to the internal volume of the larva, whereas smaller volumes are prone to disproportionate errors in delivery amounts. For injections, larvae should be put on ice, or in refrigeration at 4 °C, for 15 minutes to anaesthetize the larvae.

**Background microbiome.** The tobacco hornworm microbiome is thought to be transient, and not residential [32,33]. That is, any bacterium present is presumed to come from food and outside sources. The gut tract has a high pH and is thought to be too hostile to foster a consistent residential microbiome [33]. Previous work has shown that larvae raised on an artificial diet contain mostly gram-positive cocci and coryneforms, including *Staphylococcus*, *Pediococcus*,

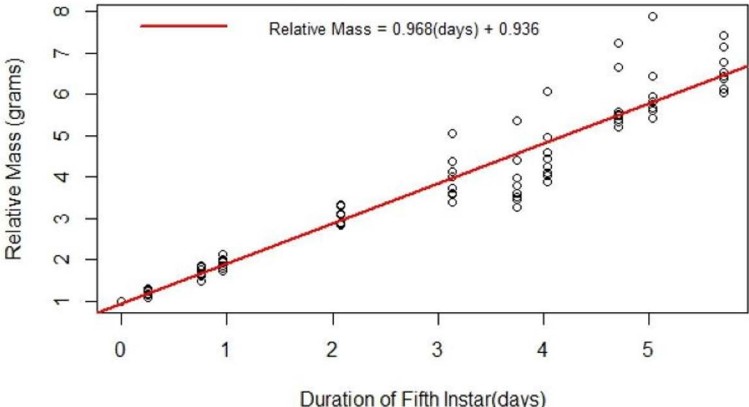

**Fig 4. The development of fifth instar larvae over time in days.** The masses of ten fifth instar larvae were recorded at varying points across time. The red line is a regression of the relative mass of fifth instar larvae throughout time in days. The relative mass is the mass of the worm at the time point it was weighed at divided by the initial mass.

**Table 4. Total volume of hemolymph drawn serially from hornworm larvae.** The volume of hemolymph was drawn five times from each of nine fifth instar larva. The beginning and final masses are the darkened columns. Two of the nine hornworm larvae died after the third draw, Hornworm 2 and Hornworm 8. The death is represented by the X in the 'Final Mass (grams)' column. Hornworm 6 only had two hemolymph draws to allow for the hornworm to recover and grow to a more robust weight. The zeros in the 'Number of Hemolymph Draws' represent that no hemolymph was drawn during that number of the draws. Hornworm 2 and Hornworm 8 have zeros following the third hemolymph draw due to their death. The total volume of hemolymph that was drawn is shown in the far-right column.

Volume of Hemolymph Drawn (µL)

| Hornworm | Beginning Mass (grams) | Number of Hemolymph Draws | | | | | Final Mass (grams) | Total Volume Hemolymph Drawn (µL) |
|---|---|---|---|---|---|---|---|---|
| | | 1st | 2nd | 3rd | 4th | 5th | | |
| 1 | 1.16 | 40 | 20 | 150 | 50 | 80 | 8.20 | 340 |
| 2 | 1.24 | 12 | 320 | 120 | 0 | 0 | X | 452 |
| 3 | 1.16 | 25 | 45 | 260 | 35 | 120 | 9.78 | 485 |
| 4 | 1.16 | 25 | 62 | 120 | 95 | 380 | 9.12 | 682 |
| 5 | 1.73 | 32 | 150 | 45 | 85 | 40 | 9.75 | 352 |
| 6 | 1.67 | 110 | 0 | 0 | 0 | 340 | 2.12 | 450 |
| 7 | 2.04 | 30 | 70 | 180 | 45 | 100 | 8.06 | 425 |
| 8 | 1.59 | 0 | 95 | 32 | 0 | 0 | X | 130 |
| 9 | 1.93 | 35 | 330 | 120 | 115 | 25 | 6.00 | 625 |

*Micrococcus*, and *Corynebacterium* [34]. The background microbiome is potentially an important factor when considering the use of tobacco hornworms in experiments involving virulent pathogens.

We determined the microbiome of a sample group of larvae from our colony. Each of five individuals were surface sterilized with 70% ethanol, anesthetized by chilling on ice for 15 minutes before their heads were removed and the body was suspended in a separate tube with phosphate buffer solution (PBS) which was then homogenized with a Tissue-Tearor™. The homogenates were sonicated for 45 seconds, then serially diluted into PBS and plated on the following media: Luria-Bertani (LB), Tryptic Soy Agar (TSA), De Man-Rogosa-Sharpe Agar (MRS), Brain-Heart Infusion (BHI), and Columbia Blood Agar (CBA). Plates were incubated at 37 °C for 24 hours before being analyzed.

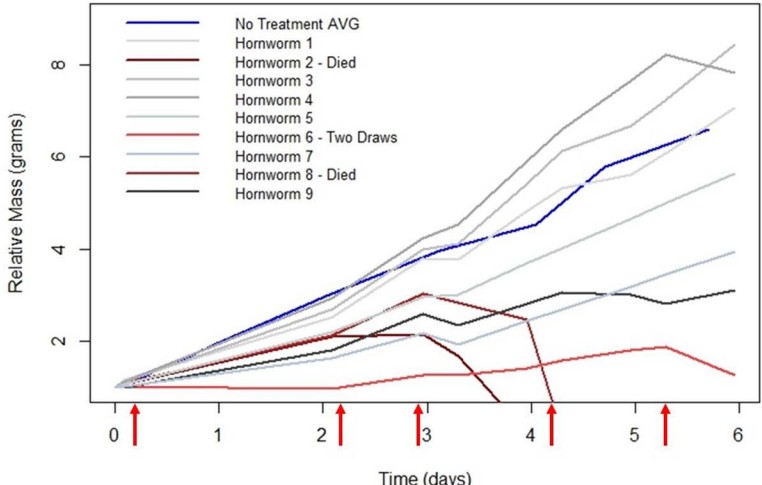

**Fig 5. The relative mass (grams) of hornworms across time with serial hemolymph draws.** The relative mass (grams) of nine hornworms that had hemolymph serially drawn five times. With the exception of the blue control line labeled 'No Treatment AVG', each line represents one larva. The blue line shows the no treatment control of the average relative mass over time. The two hornworms that perished after the third draw (Hornworm 2 & 8) are differing shades of dark red and their death is indicated by the relative mass going to zero. Hornworm 6 is represented by a light red line, and only had hemolymph drawn twice due to the low relative mass throughout the experiment. The red arrows along the x-axis represent the time points at which the hemolymph was drawn.

Colonies were selected according to differing morphology, then streaked on a fresh plate and later isolated as a single colony. Colonies were suspended in DNA-free water and boiled at 95 °C for 15 minutes before being used in a PCR reaction with the universal bacterial primers (27F) (AGA GTT TGA TCM TCG CTC AG) and (907R) (CCG TCA ATT CMT TTR AGT T) for sequencing of the amplicons allowed bacterial identification. The goal in this study was to identify species present, not abundance or total composition of the microbiome. We found the following five species present in the sample of our overall hornworm population: *Enterococcus faecalis, Enterococcus saccharolyticus, Micrococcus luteus, Bacillus megaterium,* and *Serratia marcescens.* Our results match reports [32], with ours showing predominantly gram-positive cocci species. The two outliers of this group are *Bacillus megaterium* and *Serratia marcescens*, but both bacterial species are known to reside in diverse environments.

The background microbiome may be an important factor when conducting bacterial or fungal virulence assays. When injecting bacteria into the hornworm larvae, a selective media for a specific species can be used when plating dissected and homogenized tissue to prevent bacterial species present in the background microbiome from affecting bacterial counts. For example, when injecting a gram-negative species such as *Escherichia coli*, a selective medium such as MacConkey agar can be used to isolate and differentiate gram-negative bacteria that are lactose fermenters [35].

**Prepupal stage.** Once fifth instar larvae cease feeding, they begin a five-to-seven-day prepupal stage which includes the following:

- Fecal pellet coating (Fig 6): at the end of the feeding phase of the fifth instar, larvae produce fecal pellets that are coated with uric acid, giving it a chalky appearance - 'frosted frass' [36, 37]. The coated fecal pellet indicates the end of the fifth instar feeding and the beginning of the prepupal phase.

- Body wetting (Fig 7ABC): the larva 'grooms' itself and covers its body with a regurgitated fluid. Fluids are also defecated. This phenomenon is responsible for a large drop in weight from the end of the fifth instar to the prepupal phase

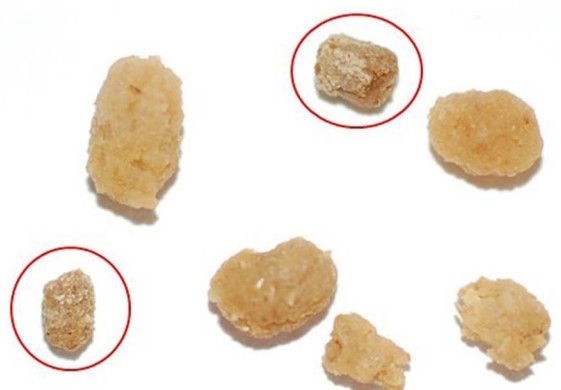

**Fig 6. Coated fecal pellets.** The coated fecal pellets are also known as 'frosted frass'. The coating is uric acid and signals the cessation of feeding in the fifth instar and the beginning of the prepupal phase. The figure shows two frosted frass that are circled in red, as compared to normal frass.

(Fig 8A). The term 'body wetting' was first described Baumhover et al. 1977, and the term was coined by Reinecke, Buckner, and Grugel [31,38]

- Peristalsis, wandering: following the body wetting phase, prepupal larvae have abdominal peristaltic waves and pulsations before they begin to wander and move substantially as if searching for a place to pupate [39].

- Dorsal pigmentation (Fig 8A): a prominent stripe that is purplish, or a dark blue that pulsates on the dorsal surface.

- Metathoracic bars (Fig 8BC): once the wandering phase is over, the prepupal larvae will remain stationary and develop dark bar-like pigmentation along the side of their bodies which signals the end of the prepupal stage and the beginning of pupation [31].

The prepupal stage typically lasts on average five-to-seven days, depending on the temperature (Fig 7D). At 22–25 °C, the prepupal stage can last six-to-eight days, whereas at 26–30 °C the prepupal stage lasts three-to-six days.

**Pupal stage.** Once pupal ecdysis begins, the process of shedding the larval cuticle and the full development of the pupa takes only 150 minutes [40]. Once the larval skin is shed off, the pupa develops from a soft turquoise and green color to a tanned and hardened outside surface in a biological process called sclerotization. The pupal ecdysis process can be observed in Fig 9. The pupae will remain stationary under the substrate, unless prodded, while metamorphosis takes place. There are numerous previous studies that have discovered and monitored the physiological changes throughout pupal development and metamorphosis. However, for the importance of this work, we determined that under the laboratory conditions of 28 °C, 40–60% relative humidity, and a photoperiod of 8L:16D, the pupal phase lasts two-to-five weeks until adult eclosion.

During this life phase, the pupae can be sexed to determine whether the emerging moth is male or female. Males have two small bumps on the ventral tip of the abdomen on segment nine, which is the next to last segment. For females, segment nine is smooth, but there is a thin groove on segment eight [41].

**Adults.** The emergence of the adult from its pupal case occurs with the wings shriveled and wet; the wings can stretch. As the adult climbs from its pupal substrate, the wings expand over a period of hours. Gravity appears to assist the orientation of the wings, so it is important that a suitable, vertical climbing surface is provided. Along with the help of gravity, the venation of the wing cuticle is inflated to its full size by circulatory hemolymph pressure. This plasticity of the wings allows for an expansion that has been reported to be triggered by the eclosion hormone, and tanning hormone,

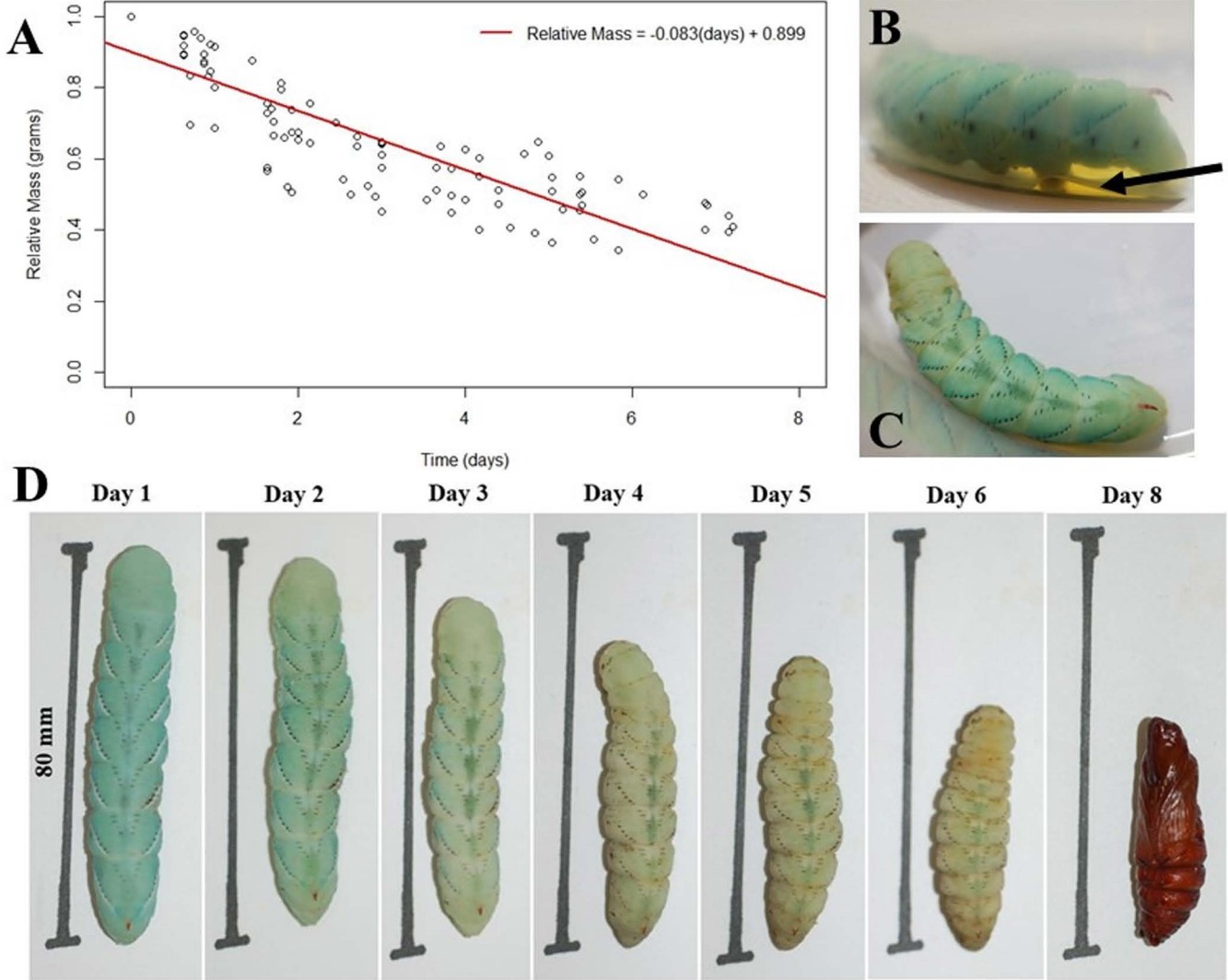

**Fig 7. Characteristics of the pre-pupal stage.** The prepupal stage. This phase is characterized by the sharp decrease in body mass from the fifth instar to the pupa. This loss of body mass is due to the larva covering their body with regurgitated fluids as well as excreting excess fluids throughout this stage. A) The relative change of mass during the prepupal phase shown in a graph format. A total of ten hornworm larvae were weighed at the end of the fifth instar phase to obtain the relative mass starting the prepupal phase. Each point represents a prepupal hornworm at various time points until pupation at day 7–8. B) Prepupal larva excreting fluid following the grooming behavior. The fluid indicated by the arrow is the amount of fluid excreted in 16 hours (0.67 days) following the beginning of the prepupal phase. C) The same body wetting prepupal larva shown from above. D) The sequence of the same single prepupal larva throughout the prepupal stage incubated at 28 °C.

bursicon, 3–4 hours prior to emergence from the pupa [42]. Following eclosion, our adults were moved from the pupation containers to the net enclosure where they were exposed to the 8L:16D photoperiod.

Previous work found that an adult composition of 50% males resulted in the highest rate of oviposition [43]. In our studies, no effort was made to adjust the sex ratio, all adults being put in the enclosure. The number of adult moths in the enclosure varied but generally ranged from 10 individuals to 30 at a time. Eggs were laid at varying times throughout the

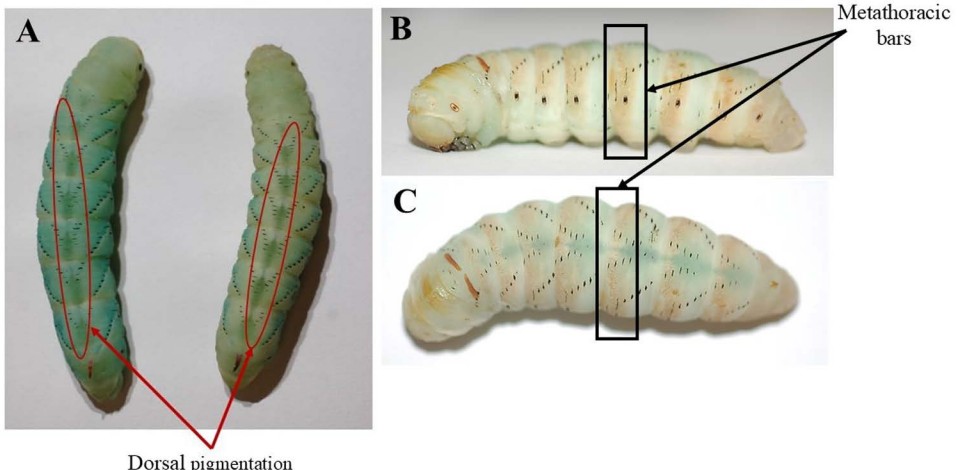

**Fig 8. Additional prepupal descriptors: dorsal pigmentation and metathoracic bars.** A) The dorsal pigmentation in the prepupal stage is circled in red. The prominent stripe pulsates from top to bottom on the dorsal slide of the larva. B & C) The metathoracic bars in the prepupal stage are highlighted by the black boxes. This stage is characterized by dark brown bar-like pigmentation along the lateral sides, as well as the top of the head. At this point in the prepupal cycle, the larvae remain stationary. The transition into pupation will occur within 48 hours of these metathoracic bars developing. B) Larva seen from lateral view. C) The same larva seen from dorsal view.

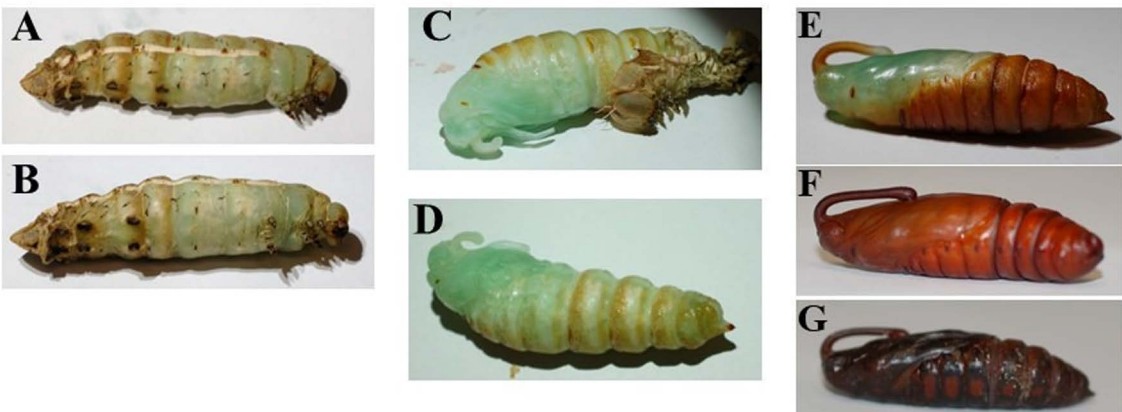

**Fig 9. Pupa throughout larval ecdysis and pupal development.** A. Lateral view of a prepupal larva that has begun the pupal ecdysis process. B. Ventral view of the same prepupal larva C. The larval skin has almost fully been shed, and the proboscis and wing tissue has shifted. D. The larval skin has fully been shed off and now the pupa will begin to sclerotize as the soft tissue tans and hardens. E. The pupa now has a hardened exterior but isn't fully tanned. F. The pupa has a fully tanned exterior and is about a week old. G. The pupa is three weeks old and is 12 hours from adult eclosion.

16-hour dark period, and for about 30–60 minutes after the onset of light. Eggs were collected daily, or every other day, typically after the first two hours of the light phase to allow for the adult activity of laying and feeding to cease. The eggs should be collected prior to day three of laying, as hatchlings may begin to emerge and will perish inside the enclosure without access to food. Depending on the number of eggs collected at a time, they were either transferred to containers at 28 °C, or to the refrigerator at 4 °C.

**Table 5. Percentage of eggs hatching after refrigeration. On the duration of time in days kept at 4 °C. Sample sizes were greater than 120 eggs for each percentage calculated.**

| Percentage of Eggs Hatching at 25–30 °C after Refrigeration (4 °C) | |
|---|---|
| Days in the Fridge | Percent of Eggs Hatched |
| 10 | 20.6 |
| 7 | 41.7 |
| 5 | 75.0 |
| 4 | 82.1 |
| 0 | 90.7 |

### Survival of manipulations used to synchronize larvae

At laying, eggs are a light turquoise color. If fertile, the eggs turn yellow and hatch in three-to-five days at 25–28 °C. In our population, approximately 10–15% of eggs laid did not hatch, presumably due to infertility. On average about 20% of each brood perished between the first and fifth instar.

Eggs can be stored at 4 °C for seven-to-ten days with some detriment to hatching and mortality (Table 5). However, eggs are often produced in abundance, and temporary refrigeration allows the synchronized hatching of legs laid up to a week apart. Eggs will hatch in three to five days when held at 25–30 °C. Fifth instar larvae can also be stored in the refrigerator (4 °C) for three-to-four days to delay growth and development to synchronize experiments, though there are mortality consequences for storing larvae at this temperature (Table 6).

### Larval color is affected by diet

As has been reported before, the color of larvae depends on diet. Wildtype larvae fed on tomato leaves were green, whereas larvae fed on artificial diet were a blue or blue-green color. The green coloration has been shown to be due to the presence of both a blue bile pigment protein and a yellow carotenoid protein, with the combination complex called insectoverdin [44–47].

The blue coloration is just the blue bile pigment complex being expressed without the presence of the plant material in the diet [47].

In addition to the diet's effect, larval color is apparently affected by physiological factors. Consistent with previous reports, we observed various color morphs. Examples of different coloration morphology observed in our colonies can be observed in Fig 10. There are two well characterized mutant phenotypes: a black and white morphology in *Manduca sexta* larvae. We observed the black morphology in our laboratory colony (Fig 10G). The white morphology is caused by the combination of a reduction in insecticyanin production in the epidermis and high titers of juvenile hormone (JH) at the beginning of the first instar [47]. The black morphology is due to the single sex-linked gene and the differing expression

**Table 6. Percent survival of fifth instar after refrigeration. Percent survival of fifth instar larvae at 4 °C, depending on the duration of time the larvae spent in the refrigerator. The sample size for fifth instar larvae held at refrigeration temperature was 15–25 individuals for each percentage calculated.**

| Percent Survival of Fifth Instar Larvae at 4 °C | |
|---|---|
| Days at 4 °C | Percent Survival (%) |
| 1 | 98.5 |
| 2 | 96.1 |
| 3 | 92.3 |
| 4 | 85 |

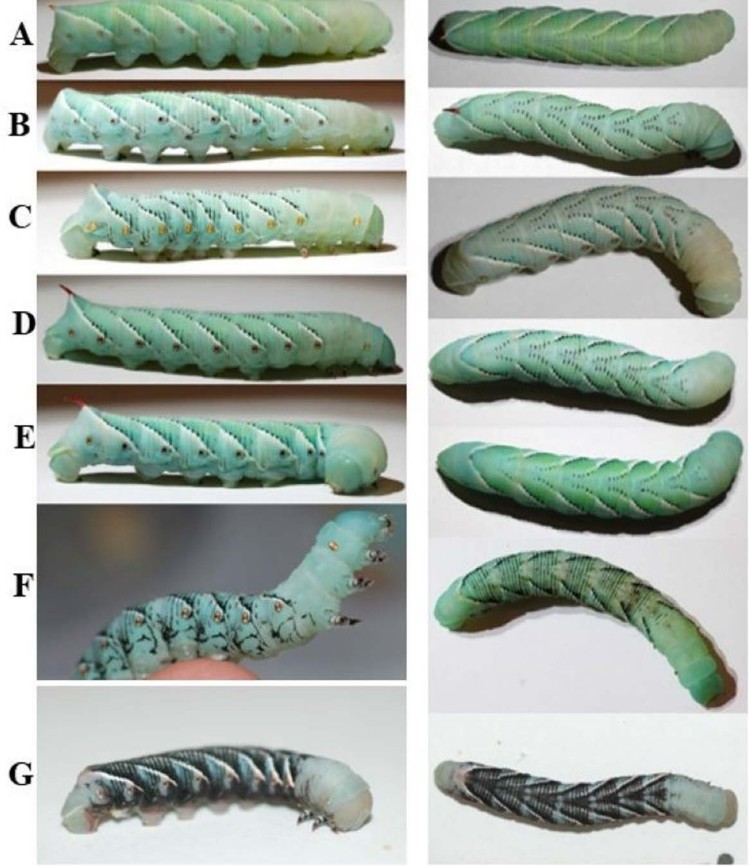

**Fig 10. Color morphs of hornworms seen in the laboratory colony.** The left column shows a fifth instar larva from the lateral side. The right column shows the same fifth instar larva from above. There is a variety of ratios of white and black striping. **G)** shows an example of a black color morph mutant.

and production levels of JH during the molting cycle [48]. This phenotype is seen in wildtype diet reared larvae in which a polyphenism of the black color morph can be induced by exposing the larvae to heat stress or a neck/abdominal ligature [49].

## Discussion

Invertebrates provide practical and ethical alternatives to the use of vertebrates as model organisms in biology. In the last decade, the greater wax moth (*Galleria mellonella*) has experienced increased use, especially as an infection model for bacteria and fungi [10,50,51]. Wax moths are easy to obtain in large numbers, but they are moderately small and thus do not allow many types of manipulation that would be useful for infection studies and other investigations.

For many purposes, the tobacco hornworm offers a close parallel to the wax moth, with the advantage of a larger size, commonly ten grams as a larva. In addition to easier handling, this size makes them a better system for measuring quantitative responses to an infection, such as a change in mass over time. Like the wax moth, hornworms are available commercially. But commercially sourced larvae of either moth are of variable quality for research and often lack uniformity; shipping may further expose them to extreme temperatures and other stressors [49, 50]. To ensure uniformity, some laboratories rear their own waxworms [52–59]. We suggest it is practical to raise hornworms.

Our paper offers a practical protocol for rearing the tobacco hornworm in a laboratory. Previous publications have described the rearing and use of the tobacco hornworm for physiological studies [29,31,60], but those protocols omit many details and are not easily followed. The goal here is to augment and simplify protocols so that any lab with no experience might rear them.

Establishing and maintaining a breeding laboratory population of the tobacco hornworms is easy and inexpensive. Egg-to-egg generation time is just over a month. The initial setup cost for the equipment necessary for this protocol is minimal, and the cost for feeding and maintaining a colony of about 500 individual larvae is about $100 per month plus technician time. The time commitment to maintain a colony of this size is roughly 15–20 hours a week. Purchased larvae through suppliers may be cheaper when considering technician time and initial set-up cost, the inability to standardize the sizing and instar age of the larvae, along with the stress of shipping makes use of purchased larvae questionable when attempting to maximize reproducibility.

The larval development is consistent in both time and size, the growth rate being determined by temperature and humidity. Fifth instar larvae are large, growing from about 1 gram to 10+ grams in 4–5-days. This size is equivalent to a small laboratory mouse and allows for easy handling as well as many types of assays not possible with waxworms – some of which we report here. Hornworm larvae are robust as they are routinely washed with 70% ethanol with no ill effect. The egg to adult survival in our laboratory colony is observed to be about 80%. A disadvantage to hornworm husbandry is that larvae need to be housed individually due to larva-larva cannibalism.

## Supporting information

**S1 Dataset. Hornworm Larvae Length and Mass Data.** This dataset is what Figure 3 is built from. Each data point from the dataset shows one individual hornworm that was weighed and had its length measured at a random point in the life cycle. A total of 38 hornworms were measured for this dataset.
(CSV)

**S2 Dataset. Hornworm Ratio of Growth from Beginning of 5th Instar.** This dataset is what Figure 4 is built from. Eight hornworms were weighed immediately following the skin shed from the fourth instar, which signaled the beginning of the fifth instar. Each hornworm was weighed at various timepoints and the weight recorded was divided by the initial weight at time point zero. This shows the growth ratio over time on Fig 4.
(CSV)

**S3 Dataset. Hemolymph Drawn Serially Datasets.** These datasets are what Figure 5 (Fig. 5) and Table 4 are built from. Nine hornworms were selected as healthy early fifth instar larvae for their hemolymph to be drawn serially over the course of five days. Each hornworm was weighed at various time points at and between the hemolymph drawings, and the growth ratio was determined by dividing the weight at the time point by the original starting weight (at time zero). This growth was then compared to the growth of healthy hornworms in the fifth instar that were not handled beyond weighing (No Treatment Control), which is depicted in Fig 4.
(CSV)

**S4 Dataset. Prepupal Loss of Mass Dataset.** This dataset shows the loss of mass that individual hornworm larvae undergo during their prepupal phase in the life cycle. Ten individual larvae were selected and weighed at the end of their fifth instar, which was depicted as 'Starting Mass'. Each individual was then weighed at various time points over the course of five to eight days, and the weight was divided by the original starting mass to determine the ratio of relative mass lost throughout this phase. These data are depicted in Figure 7A.
(CSV)

## Acknowledgments

The authors would like to thank Dr. Holly Wichman, and all of those involved with the Wichman-Miller Laboratory group at the University of Idaho for their advice, support, and expertise.

## Author contributions

**Conceptualization:** Emma Kay Spencer, James J. Bull.

**Data curation:** Emma Kay Spencer.

**Funding acquisition:** Craig R. Miller.

**Investigation:** Emma Kay Spencer.

**Methodology:** Emma Kay Spencer.

**Supervision:** Craig R. Miller, James J. Bull.

**Visualization:** Emma Kay Spencer.

**Writing – original draft:** Emma Kay Spencer.

**Writing – review & editing:** Emma Kay Spencer, Craig R Miller, James J. Bull.

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
