## [Decision Letter · Decision Letter 0]

4 Feb 2025

PONE-D-24-57766Standardized methods for rearing a moth larva, Manduca sexta, in a laboratory settingPLOS ONE

Dear Dr. Spencer,

Thank you for submitting your manuscript to PLOS ONE. After careful consideration, we feel that it has merit but does not fully meet PLOS ONE’s publication criteria as it currently stands. Therefore, we invite you to submit a revised version of the manuscript that addresses the points raised during the review process.

We look forward to receiving your revised manuscript.

Kind regards,

Shouke Zhang

Academic Editor

PLOS ONE

2. Thank you for stating the following financial disclosure:  [Research reported in this publication was supported by the National Institute Of General Medical Sciences of the National Institutes of Health under Award Number P20GM104420. The content is solely the responsibility of the authors and does not necessarily represent the official views of the National Institutes of Health.]. 

4. Please include a caption for figure 1 and 13.

Additional Editor Comments (if provided):

Reviewers' comments:

Reviewer's Responses to Questions

**Comments to the Author**

1. Is the manuscript technically sound, and do the data support the conclusions?

Reviewer #1: Yes

Reviewer #2: Yes

2. Has the statistical analysis been performed appropriately and rigorously? 

Reviewer #1: Yes

Reviewer #2: Yes

3. Have the authors made all data underlying the findings in their manuscript fully available?

Reviewer #1: Yes

Reviewer #2: Yes

4. Is the manuscript presented in an intelligible fashion and written in standard English?

Reviewer #1: Yes

Reviewer #2: Yes

5. Review Comments to the Author

Reviewer #1: The study reported a method about how to rear a moth larva. I have some suggestions and concerns that need to be addressed.

Lines 82-83, references are needed here.

Lines 82-88, in addition to the aspects the authors mentioned, there are other factors that are important for the suitability of an insect is set as a research model, such as, genetic background information. Consequently, whether genome or transcriptome information is available

Meanwhile, the life history, host plants, ecological or economic significance also needs a brief introduction.

Line 102, is “great lakes” a geolocational place? Geolocational information are needed.

Line 101-108, if these insects were not obtained from the wild, how can the authors ensure that the results of the following research reflect the actual situation in the field?

Table 1 italicize Drosophila.

Lines 139-195, this section would be more appropriate in the M&M section.

Table 2 and 3, revise it into a three-line table.

Table 4, how many replicates were included in each treatment?

Line 271-274, what is the volume of injection that would not cause bleeding?

Line 282, references are needed for the claim.

Lines 284-286, italicize the genus names.

Lines 299-300, why the authors used 27F-907R instead of 27F-1492R?

Lines 306-312, references are needed in this section.

As for figures, I would suggest the authors considering merge some of them as 14 figures is too much for such a study.

Reviewer #2: This MS described a protocol for rearing a kind of larger size larva, tobacco hornworm, Manduca sexta, which is useful in physiological studies. The results showed the methods were easily done, effective and cheaper, but a few issues remain.

1) in Introduction section, the authors mentioned the tobacco hornworm larvae could be purchased, more details about the disadvantages of these larvae including price and quality were needed to highlight the importance of this research, I think.

2) Line 124-125, the material composition of pre-mixed dry food is necessary for larval rearing.

3) Line 147, the best humidity at each rearing stages is need to be given.

4) Line 152, the suitable number of larvae reared simultaneously in per flask should be known. It’s important to avoid cannibalism.

5) Line 209-213, the duration of each stages should be more accurate. For example, pupal duration could span 2-5 weeks, it’s a significant difference. Foods or the rearing conditions cause that?

6) It would be nice if the study methods in ‘Biology of the life cycle’, ‘Fifth instar…’ and ‘Microbiome’ sections could be transferred to the ‘Materials and Methods’section.

7) I think it’s too simple to described the background microbiome as Table 5.

8) The authors mentioned the rearing costs is cheaper, but there no details for comparison with other resources like purchased larvae.

9) The pictures listed in present MS were very poor.

6. PLOS authors have the option to publish the peer review history of their article (what does this mean? ). If published, this will include your full peer review and any attached files.

**Do you want your identity to be public for this peer review?** For information about this choice, including consent withdrawal, please see our Privacy Policy .

Reviewer #1: No

Reviewer #2: No

---

## [Author Response · Author response to Decision Letter 0]

28 Feb 2025

Reviewer 1

1. Lines 40-45, references are needed here.

Author response: We have added a reference that compares the size of a mouse to a fifth instar larvae, which is reference 14.

2. Lines 40-52, in addition to the aspects the authors mentioned, there are other factors that are important for the suitability of an insect is set as a research model, such as, genetic background information. Consequently, whether genome or transcriptome information is available? Meanwhile, the life history, host plants, ecological or economic significance also needs a brief introduction

Author response: The information that you suggested we include, life history, host plants, ecological/economic significance, is relevant to this manuscript. We expanded this paragraph to include more information, and with this came additional references to support this information. The references added were 15-25.

3. Line 77, is “great lakes” a geolocational place? Geolocational information are needed.

Author response: “Great Lakes” is referring to a company name in this particular instance. The company itself is located in the state of Michigan in the United States. Locations for both companies are now given in lines 76-77

4. Line 76-84, if these insects were not obtained from the wild, how can the authors ensure that the results of the following research reflect the actual situation in the field?

Author response: This manuscript refers to rearing tobacco hornworms throughout their lifecycle in a laboratory setting to use the larvae as model organisms in a variety of studies. We are not trying to mimic any natural infections. Rather we are proposing hornworms as model for various types of pathogen studies, in the same way that mice, waxworms, and other organisms are used to study infections. Studies that used tobacco hornworms in a laboratory setting for the study of insect physiology also obtained insects from a manufactured source rather than obtain the stock directly from the wild, just as we did in our study. We added lines 81-84 to address this comment.

5. Table 1 italicize Drosophila

Author response: Done.

6. Lines 118-178, this section would be more appropriate in the M&M section.

Author response: We agree with this comment and have moved this section to Methods. We think that implementing this suggestion made the manuscript more cohesive and improved the flow from start to finish. This change did, however, make it less feasible to make a change suggested by the other reviewer.

7. Table 2 and 3, revise it into a three-line table

Author response: Table 2 & 3 were made into a three-line table by adding a title to the table.

8. Table 4, how many replicates were included in each treatment?

Author response: In the manuscript and captions of both Figure 5 and Table 4, we explained that the data are from one sample group of nine hornworms. The text gives the number of replicates.

9. Lines 273-275, what is the volume of injection that would not cause bleeding?

Author response: We added the answer to this question into the manuscript at line 273-275.

10. Line 280, references are needed for the claim.

Author response: We added two references that show this claim is accurate. References 32 & 33.

11. Lines 284-385, italicize the genus names.

Author response: We now italicize all genus names.

12. Lines 299-300 why the authors used 27F-907R instead of 27F-1492R?

Author response: We used the 27F – 907R primers rather than the 27F – 1429R primers because the former were already available to us, and they were sufficient. The information we sought was merely to the species level. The 27F – 907R targets a shorter, more conserved region of the 16S gene as compared to the 27F – 1429R, which are better suited for a fine scale species identification within a specific bacterial group.

13. Line 409-315, references needed in this section.

Author response: Reference 35 now added.

14. As for figures, I would suggest the authors consider merge some of them as 14 figures is too much for such a study

Author response: We agree. We have thus removed Figure 6, and combined Figures 8, 9, and 10 and combined Figures 11 & 12. This took the number of figures from 14 to 10.

Reviewer 2

1. In Introduction section, the authors mentioned the tobacco hornworm larvae could be purchased, more details about the disadvantages of these larvae including price and quality were needed to highlight the importance of this research, I think.

Author response: We didn’t mention the feasibility of buying hornworms for research because we wanted to emphasize the inexpensive nature of this model organism as well as the value of rearing insects directly. However, we have now added an introductory text that mentions this alternative, lines 54-66. For this section, three additional references 26 – 28 were added.

2. Lines 99-102, the material composition of pre-mixed dry food is necessary for larval rearing.

Author response: As stated in the manuscript, the recipe from Great Lakes Hornworms is proprietary. They have since given us a brief list of ingredients, which was added in Line 101-102.

3. Line 120, the best humidity at each rearing stages is need to be given.

Author response: We added our humidity and temperature as a blanket statement at the beginning of the ‘Protocol for Rearing’ statement (Lines 120-161). All steps of the protocol are completed at this given humidity and temperature unless otherwise noted.

4. Lines 134-140, the suitable number of larvae reared simultaneously in per flask should be known. It’s important to avoid cannibalism.

Author response: A clarifying statement was added for lines 135-136 to make it clear that they are held individually in each flask. They otherwise eat each other.

5. Lines 196-200, the duration of each stages should be more accurate. For example, pupal duration could span 2-5 weeks, it’s a significant difference. Foods or the rearing conditions cause that?

Author response: The duration of the pupal stage of 2 – 5 weeks was what we observed, and we felt it necessary to report the (rather large) variation. A similar range was also observed by other groups who have published on this species, which can be seen in the references used throughout the manuscript. We do not know the cause, as the food and rearing conditions were similar across the many individuals that have been reared throughout the duration of the program. It’s not unexpected that wild strains have (non-genetic) variation in timing of major events, as the variation could have important fitness benefits. The classic example is the bet-hedging variation of germination probability in desert annual plants.

6. It would be nice if the study methods in ‘Biology of the life cycle’, ‘Fifth instar…’ and ‘Microbiome’ sections could be transferred to the ‘Materials and Methods’ section.

Author response: We appreciate the reviewer giving careful thought to the structure of our paper. We were faced with a dilemma of the suggestion of a different restructuring from both reviewers. We opted to go with the other suggestion.

7. I think it’s too simple to describe the background microbiome as Table 5.

Author response: Our description is simple, for sure. However, our study is just a preliminary exploration into bacterial species typically found in the hornworm, and more specifically, what species were found in our hornworm population. Our goal was to know if there were bacterial species that might be confused morphologically with pathogenic bacteria that we may use in the future of a bacterial infection study. We agree that our efforts did not match the comprehensive efforts of groups studying gut microbiomes in other species. Nonetheless, we thought that it would be useful to provide this preliminary information. We eliminated Table 5 and incorporated the information into the text in lines 301-305, but we would be willing to remove even that if omission is what this reviewer thinks appropriate. The data have no real impact on our paper.

8. The authors mentioned the rearing costs is cheaper, but there no details for comparison with other resources like purchased larvae.

Author response: We should have included the cost of purchasing larvae directly in both the introduction and discussion section. As stated in the response to comment 1, we added lines 124 – 136 in the introduction. In the discussion section, we added a section, lines 598 – 602, that address this concern.

9. The pictures listed in present MS were very poor.

Author response: Thank you for letting us know. We followed the instructions given by PLOS ONE for image formatting with size but neglected to check the resolution. The images submitted were below 300 dpi, which is the minimum requirement for PLOS One. Upon resubmittal, we will make sure that the images are between 300-600 dpi.

---

## [Decision Letter · Decision Letter 1]

16 Mar 2025

Standardized methods for rearing a moth larva, Manduca sexta, in a laboratory setting

PONE-D-24-57766R1

Dear Dr. Emma Kay Spencer,

We’re pleased to inform you that your manuscript has been judged scientifically suitable for publication and will be formally accepted for publication once it meets all outstanding technical requirements.

Kind regards,

Shouke Zhang

Academic Editor

PLOS ONE

Additional Editor Comments (optional):

Reviewers' comments:

Reviewer's Responses to Questions

**Comments to the Author**

1. If the authors have adequately addressed your comments raised in a previous round of review and you feel that this manuscript is now acceptable for publication, you may indicate that here to bypass the “Comments to the Author” section, enter your conflict of interest statement in the “Confidential to Editor” section, and submit your "Accept" recommendation.

Reviewer #1: All comments have been addressed

Reviewer #2: All comments have been addressed

2. Is the manuscript technically sound, and do the data support the conclusions?

Reviewer #1: Yes

Reviewer #2: Yes

3. Has the statistical analysis been performed appropriately and rigorously? 

Reviewer #1: Yes

Reviewer #2: Yes

4. Have the authors made all data underlying the findings in their manuscript fully available?

Reviewer #1: Yes

Reviewer #2: Yes

5. Is the manuscript presented in an intelligible fashion and written in standard English?

Reviewer #1: Yes

Reviewer #2: Yes

6. Review Comments to the Author

Reviewer #1: The authors have successfully addressed my previous concerns. I have no further comments.

Reviewer #2: This study described a easily done, effective and cheaper protocol for rearing a kind of larger size larva, tobacco hornworm, Manduca sexta, which is useful in physiological studies. All comments have been addressed. No more comments.

7. PLOS authors have the option to publish the peer review history of their article (what does this mean? ). If published, this will include your full peer review and any attached files.

**Do you want your identity to be public for this peer review?** For information about this choice, including consent withdrawal, please see our Privacy Policy .

Reviewer #1: No

Reviewer #2: **Yes: ** Shu Jinping

---

## [Editor Report · Acceptance letter]

PONE-D-24-57766R1

PLOS ONE

Dear Dr. Spencer,

I'm pleased to inform you that your manuscript has been deemed suitable for publication in PLOS ONE. Congratulations! Your manuscript is now being handed over to our production team.

Kind regards,

on behalf of

Dr. Shouke Zhang

Academic Editor

PLOS ONE